# Shift Work Including Night Work and Long Working Hours in Industrial Plants Increases the Risk of Atherosclerosis

**DOI:** 10.3390/ijerph16030521

**Published:** 2019-02-12

**Authors:** Marit Skogstad, Asgeir Mamen, Lars-Kristian Lunde, Bente Ulvestad, Dagfinn Matre, Hans Christian D. Aass, Reidun Øvstebø, Pia Nielsen, Kari N. Samuelsen, Øivind Skare, Per Anton Sirnes

**Affiliations:** 1Department of Occupational Medicine and Epidemiology, National Institute of Occupational Health, Box 8149 Department, 0033 Oslo, Norway; bente.ulvestad@stami.no (B.U.); oivind.skare@stami.no (Ø.S.); 2Norwegian School of Health Sciences, Kristiania University College, Box 1190 Sentrum, 0107 Oslo, Norway; asgeir.mamen@nhck.no; 3Department of Work Psychology and Physiology, National Institute of Occupational Health, Box 8149 0033 Oslo, Norway; lars-kristian.lunde@stami.no (L.K.L.); dagfinn.matre@stami.no (D.M.); 4The Blood Cell Research Group, Department of Medical Biochemistry, Oslo University Hospital, 0450 Ullevaal, Norway; h.c.aass@medisin.uio.no (H.C.D.A.); reidun.ovstebo@medisin.uio.no (R.Ø.); 5Ringvoll BHT, 1523 Moss, Norway; pia.nielsen@ringvollbht.no (P.N.); kari.samuelsen@ringvollbht.no (K.N.S.); 6Ostlandske Hjertesenter, 1523 Moss, Norway; pas@cardio.no

**Keywords:** shift work, cardiovascular, occupational health

## Abstract

There is an abundance of literature reporting an association between shift work and cardiovascular disease (CVD). Few studies have examined early manifestation of CVD using advanced modern methodology. We established a group of 65 shift workers and 29 day workers (controls) in two industrial plants. For the shift workers, the shift schedule includes rotating shifts with day, evening and nightshifts, some day and nightshifts lasting for 12 h. The current paper describes cross-sectional data in a study running for three years. We collected background data by questionnaire and measured blood pressure, heart rate, lipids, glycosylated hemoglobin (HbA1c) and C-reactive protein (CRP). We examined arterial stiffness (central blood pressure, augmentation pressure and index, and pulse wave velocity) by the use of SphygmoCor^®^ (AtCor Medical Pty Ltd, Sydney, Australia) and the carotid arteries by ultrasound. We assessed VO_2max_ by bicycle ergometry. We applied linear and logistic regression to evaluate associations between total number of years in shift work and cardiovascular outcome measures. The day workers were older and had more pronounced arterial stiffness compared to the shift workers. Number of years as a shift worker was associated with increased carotid intima media thickness (max IMT) (B = 0.015, *p* = 0.009) and an elevated CRP (B = 0.06, *p* = 0.03). Within the normal range for this age group, VO_2max_ was 41 (9) ml/kg/min. Rotating shift work including day and night shifts lasting up to 12 h and evening shifts are associated with CVD-risk factors. This could imply an increased risk for coronary heart disease and stroke among these workers. Therefore, preventive measures should be considered for these groups of workers in order to prevent such diseases.

## 1. Introduction

Economic globalization influences the labor market and work organization, resulting in more use of long working hours and night shifts [1]. Shift work increases the risk of workplace accidents, depression of the immune system, type II diabetes, weight gain, some cancers, anxiety, change in alcohol consumption, sleep disturbances, musculoskeletal and gastrointestinal complaints [2,3]. As for shift work and ischemic heart disease, a moderate positive association between unspecified shift work and morbidity has been reported [4]. A systematic review and meta-analysis encompassing 173,000 individuals reports an increased risk for cardiovascular disease (CVD) among shift workers compared to day workers, increasing by 7.1% for every additional five years of exposure after the first five years as shift worker [5].

CVD biological risk factors in shift workers could be due to an acute sympathoadrenal arousal changing levels of cholesterol, uric acid, glucose and potassium due to unfavorable catecholamine excretion [6]. Furthermore, night shift work could be associated with an increased risk of systemic inflammation [7,8,9].

There is a lack of studies that use ultrasound to examine the carotid artery in occupational medicine. This is also the case for arterial stiffness. Ultrasound examination of the carotid arteries is an easily accessible tool by which one can obtain the carotid intima media thickness (cIMT), and additionally, one can access early subclinical atherosclerosis by the detection of plaques. cIMT is related to blood pressure, diabetes, hypercholesterolemia, and other risk factors for atherosclerosis [10]. We are not aware of any study describing the relation between strenuous shift work schedules and early atherosclerosis. Furthermore, recent studies have shown that central blood pressure (BP), such as aortic BP estimated by pulse wave analysis, correlates better with cardiovascular events than brachial BP [11]. Arterial stiffness, demonstrated by pulse wave velocity (PWV), indicates an increased risk of CVD and all-cause mortality [12]. Thus, there is a need for studies of early manifestation of CVD and shift work including long working hours and night shifts.

In the present study, we examined blood parameters including lipids (cholesterol, low density lipoprotein (LDL) and high density lipoprotein (HDL)), glycosylated haemoglobin (HbA1c), C-reactive protein (CRP), brachial blood pressure and central blood pressure, augmentation pressure and index, pulse wave velocity, and resting heart rate by SphygmoCor^® ^ (AtCor Medical Pty Ltd, Sydney, Australia), aerobic fitness, and ultrasound measurement of the carotid arteries. Our hypothesis is that shift work, including long working hours and night shifts, gives an increased risk of early manifestations of CVD. With this study, we aimed to present the baseline results from a cohort study running for three years among industrial shift workers.

## 2. Methods

### 2.1. Study Design and Population

We recruited participants from two plants (A + B), producing insulation material in Eastern Norway, to the present prospective study. Information about the study was presented to key employees and employers along with representatives from the trade unions at both plants. Afterwards, the information about the study, with an invitation to participate, was spread through the plants’ websites and orally passed on to the workers by key personnel at both plants. We invited all eligible shift workers at the two plants to participate in the study. From plant A we recruited 42 out of 51 shift workers (82%) and from plant B we recruited 23 of 55 shift workers (42%). From plant B we also recruited 29 day workers (75 eligible), which amounts to 39% of all day workers at this plant. Thirteen of the 94 workers were female (14%) (Table 1). We tested the participants during daytime, mostly during dayshift or when they were off duty. All tests, except for the examination at the cardiologist, were performed at the occupational health services. The occupational health services and cardiologist were situated in the same building, a 45 min drive from plant A and a 5 min drive for those coming from Plant B. Altogether, the tests took between 45 and 60 min per person.

### 2.2. Shift Work Exposure

The shift work follows a “5-shift plan” schedule in a clockwise rotation (Table 2). After consecutive night shifts, the schedule includes two or three days off. Standard shift length is eight hours. In the 5-week schedule Plant A is scheduled to work seven day shifts, of which two last for 12 h; five evening shifts; and seven nightshifts, of which two last for 12 h. For plant B, the shift schedule is similar: They also work seven night shifts, in which four are consecutive shifts of 8h duration and three of 12 h duration. This group of workers has seven day shifts, of which three are of 12 h duration. The control group from Plant B works five consecutive day shifts every week. Participants were asked for the number of years of shift work. Most of the day workers (21 out of 29) had previously been employed as shift workers.

### 2.3. Exposure to Toxins

The production/packing is automatized so most of the work of the participants is sedentary, in which they supervise the production by watching big screens/computers. There could be exposures of dust—even silicate—but these are seldom. Most potential dust exposure is of big particles that would deposit in the upper airways. These potential exposures are similar in the two plants. There could be exposures to aerosols, such as ammonia, but this gas dissolves mostly in the upper airways. The potential exposures are also present for those who only work during daytime. According to the occupational health services, these exposures are of low level and are not considered to cause increased risk for CVD.

### 2.4. Medical History

We collected background data and medical history by questionnaire. All participants provided the height themselves; if in doubt, we measured it using a Seca device (Vogel & Halke, Hamburg, Germany). We measured weight in kg using a Seca 22089 balance (Hamburg, Germany). We assessed physical activity (PA) by the question: “How often do you normally exercise?”, with the response alternatives of “never, less than once per week”, “once per week”, “two to three times per week”, or “almost every day” [13]. Furthermore, we asked participants to report, in minutes per week, the total amount they were engaged in PA with a low/moderate intensity (e.g., walking) and a high degree of intensity (e.g., running, spinning).

### 2.5. Brachial Blood Pressure (BP) and Resting Heart Rate (RHR) 

While the participant was sitting, we measured blood pressure and resting heart rate (RHR), after five minutes of rest, on the left arm three times in intervals of one minute. We used the lowest values of the systolic (sBP) and the diastolic pressure (dBP) in the statistical analysis. Blood pressure and RHR were measured with BpTRU^®^ (Bp TRU medical devices, Coquitlam, BC, Canada).

### 2.6. Arterial Stiffness

We assessed central blood pressure, augmentation pressure, augmentation index (the ratio of the augmentation pressure to the central pulse pressure, (A1x and A1x75)), and pulse wave velocity (PWV) by SphygmoCor XCEL^®^ (AtCor Medical Pty Ltd., Sydney, Australia) while the participant was in supine position. We performed the measurements according to the manufacturer’s recommendations (www.atcormedical.com).

### 2.7. Carotid Artery Examination

Ultrasound examination of both carotid arteries was performed by one experienced investigator (PAS) using a Vivid E95™ Ultrasound system (GE Vingmed Ultrasound AS, Horten, Norway) with a linear 11-L D transducer. Images were stored on a separate computer and analyzed with the vascular carotid software package of EchoPAC™ version 2.02 (GE Vingmed Ultrasound AS, Horten, Norway). Frames with a good resolution and a perpendicular alignment were chosen and analyzed at the ECG p-wave. The carotid intima media thickness (cIMT) was measured over the distal 5–10 mm of the far wall of the right and left common carotid arteries and 5–8 measurements were made. All cIMT measurements from both sides of the carotid arteries were averaged to obtain an average cIMT. In addition, maximal measured IMT (max IMT) and plaque thickness from the common carotid, the carotid bulb and internal carotid artery, was recorded [14]. Carotid plaque was defined as a focal structure encroaching into the arterial lumen of at least 0.5 mm; or 50% of the surrounding IMT value; or demonstrating a thickness > 1.5 mm, as measured from the media–adventitia interface to the intima–lumen interface [15]. Assessment of stenosis and plaque in both carotid arteries was done by scanning in longitudinal and cross section from the proximal common carotid artery into the distal internal carotid artery. The carotid plaque score (CPS) was defined as the sum of the maximal plaque thickness of each of the four segments of the carotid arteries of both sides [14]. A carotid stenosis was defined as ≥50% luminal narrowing according to the NASCET criteria [16].

### 2.8. Aerobic Fitness was Determined by a Direct VO_2max_ Test

We tested participants using a graded exercise test on a cycle ergometer (Monark 874E, Monark Exercise AB, Vansbro, Sweden). The starting load was 70 W with a cadence of 70 revolutions per minute (RPM). The subjects were instructed to maintain a cadence of between 68 and 72 RPM during the test. Every minute, the resistance was increased by 28 W (0.4 Kg), until voluntary exhaustion or if the cadence dropped below 65 RPM in spite of verbal encouragement to increase it. The subject wore a heart rate belt and sender from Polar (Polar OY, Kempele, Finland). The signal from this sender was presented on the bike’s screen and recorded every minute during the test. A K5 metabolism analyzer (CosmedSrl, Rome, Italy) was attached to the test subject through a Hans Rudolph 7400 Vmask oro-nasal face mask (H R, Shawnee, KS, USA), which was fitted to the subject and then checked for leakage before the test started. Oxygen uptake was measured continuously with the K5 using the unit’s mixing chamber and 10 s measurement intervals. VO_2max_ was defined as the median of the three highest successive 10 s measurements at the end of the test. Criteria for a valid test included a respiratory exchange ratio of >1.05 or heart rate >95% of age predicted maximal heart rate, together with the test leader’s evaluation of fatigue. Time to exhaustion and end load were also recorded. The highest heart rate recorded, plus five bpm, was considered the maximal heart rate [17].

### 2.9. Blood Analyses

Ethylene diamine tetra acetic acid (EDTA) blood and whole blood (gel tubes) for serum investigations were collected for glycosylated hemoglobin (HbA1c) and lipids (cholesterol, low-density lipoprotein (LDL), high-density lipoprotein (HDL)) and C-reactive protein (CRP), respectively. The tubes were centrifuged at 30 × 1000 RPM for 10 min within 60 min of the blood being drawn from a vein. Samples were transported to the Department of Medical Biochemistry Oslo University Hospital and analyzed within 48 h. HbA1c (EDTA blood) was analyzed with a Tosoh G7 HPLC analyzer (Tosoh Bioscience, Inc. San Francisco, CA, USA) using the “high performance liquid chromatography” separation principle. The analytical variation is 1.7%. Cholesterol, LDL and HDL in serum were analyzed by enzymatic colorimetric method in the Cobas 8000. Analytical variation coefficients are 3.0%, 4.0% and 3.5%, respectively. CRP was assessed in serum by particle enhanced immune turbidimetric method on Cobas 8000 (Cobas 8000 Modular Analyzer Roche Diagnostics, www.roche.com). The analytical variation is 8.0%.

### 2.10. Ethics

The Regional Ethics Committee in Oslo approved of the study (2018/1258). We informed the participants about the study and they gave their written consent to participate. (ISRCTN42416837).

### 2.11. Statistical Analysis

We used linear regression to analyze the effect of number of years with shift work on health outcomes, except for plaque, where we applied logistic regression. Adjustment were made for age, sex and pack-years (a clinical quantification of cigarette smoking). We tested differences between all shift workers and day workers using linear and logistic regression, adjusting for age, sex and pack-years. All analyses were carried out using STATA/SE 15.1 (StataCorp LLC, Texas, USA).

## 3. Results

### 3.1. Demographic Characteristics of The Study Population

The day workers were somewhat older compared to the shift workers, 49.7 vs. 40.3 years (95%, CI: 13.6, 5.2) and reported a higher degree of weekly PA of high intensity compared to the shift workers. They had less shift work experience than the shift workers did (see Table 1). There were no differences between the two groups of shift workers (Table 1 and Table 3).

Sixteen of the workers were smokers (1 woman). Twelve of the participants reported college or university education and nine of them were day workers.

### 3.2. Blood Pressure and Resting Heart Rate 

No differences in blood pressure were detected comparing shift vs day workers (see Table 3). RHR was negatively affected by PA of high degree of intensity, but was not associated with number of years as a shift worker (see Table 4).

### 3.3. Arterial Stiffness

The day workers had higher augmentation pressure and pulse wave velocity compared with the shift workers (see Table 3). Age and pack-years were associated with increasing augmentation pressure, A1× and A1 × 75, but a high degree of physical activity had the opposite effect—protective effect (results not shown). Age and BMI were positively associated with increasing pulse wave velocity, whereas high intensity PA had a protective effect on this parameter (results not shown).

### 3.4. Carotid Artery

Plaques were equally distributed in the three groups (see Table 3). Max IMT was positively associated with number of years as shift worker (see Table 4).

### 3.5. VO_2max_

The three groups did not differ statistically in aerobic power. For plant A, VO_2max_ was 42.4 (SD = 6.2). Among shift workers and day workers; for plant B the corresponding figures were 46.6 (SD = 9.2) and 40.5 (SD = 11.3), respectively. These results are above 40 mL/Kg/min in all groups. These levels coincide with typical levels for the general Norwegian population of similar age [18].

### 3.6. Blood Analysis

There were no differences found comparing lipids, HbA1c, and CRP between the three groups. The cholesterol/HDL ratio was more favorable among those who reported a high degree of PA (results not shown). HbA1c increased with increasing age (results not shown). As for CRP, the number of years as shift worker was associated with this outcome (Table 4). This is also the case for BMI (results not shown).

## 4. Discussion

With this study of workers in industry, we have found that shift work is associated with early manifestations of CVD. Carotid max IMT and increasing CRP among shift workers are related to number of years as shift worker.

We found no effect of shift work on blood pressure (BP) parameters, but as shown in the literature [19], age was associated with increasing BP in the present study.

An association between arterial stiffness and cumulative numbers of night shifts has been described [20,21]. We found that arterial stiffness in general was associated with increasing age and increasing BMI. Physical activity, however, has a protective effect on stiffness of the arteries, demonstrated in this study by favorable results of augmentation pressure, augmentation index and pulse wave velocity. The augmentation index decreases and pulse wave velocity improves through aerobic exercise among patients with CVD, as shown in an extensive meta-analysis [22]. High intensity PA, such as running, cycling and swimming, reduces the stiffness of the arteries through vasodilatory NO-release and diminishes oxidative stress and inflammation [23].

In line with figures from our study, carotid plaques are found in 20% of individuals with LDL levels considered to be within a normal range [24]. Our study shows an association between increasing IMT in the carotid artery and number of years of shift work. This is also described in a population-based study from Finland, which reports that shift work, including regular evening and night shifts, is associated with increased IMT and carotid plaques at an early age [25]. Other studies have found that working long hours are associated with intima-media thickness [26].

A low Cholesterol/HDL ratio was associated with regular high weekly PA in the present cohort. We found no association between increased Cholesterol /HDL ratio and number of years with shift work.

In recent years, there has been an increased attention to inflammation as an independent cardiovascular risk factor besides the traditional ones, and targeting the IL-6 pathway has been shown to reduce mortality and morbidity [27]. A low CRP is beneficial to cardio vascular health, since there is an increased vascular risk with a high level of this biomarker [28]. We found that the number of years as a shift worker was associated with increasing levels of this outcome variable. CRP levels exceeding 3 mg/L seem to be linked to the presence of carotid plaques but might not be the causal factor for this early manifestation of CVD according to Eltoft et al. [29].

A strength of the present paper is that we have studied early manifestation of CVD, examining many parameters and using advanced modern methodology. Self-selection of fit, highly motivated shift workers could not be ruled out, particularly in plant B where less than 50% of the total group of shift workers were recruited. However, they did not differ from the other group of shift workers at plant A, where more than 80% of the total number of workers attended. Another strength is the available information on the number of years of shift work. This may constitute a more accurate exposure metric than dividing subjects into groups based on the current shift work status. A limitation of the study is its cross-sectional design, since caution should be taken indicating a relationship between cause and effect. This design is also subject to bias, and in the present study measures of arterial stiffness were more present among day workers, which could be interpreted as a healthy worker effect among the shift workers. Another limitation of the study is that covariates related to CVD, such as unhealthy eating habits and alcohol intake, were not measured. As for alcohol intake habits, rotating shift workers could consume less alcohol than day workers [30]. Alcohol habits are sensitive information and we decided not to ask about these, since we wanted to establish a close relationship with the workers. We will follow the present cohort for three years and through a comprehensive and a novel set of methods, such as ultrasound of the carotid artery and assessment of arterial stiffness, further elucidate the present outcomes associated with shift work.

## 5. Conclusions

In this cross-sectional study of workers in industry, the number of years of shift work was associated with early manifestations of CVD, with an increased IMT in the carotid artery and increasing CRP. This could imply an increased risk for coronary heart disease and stroke among these workers. We will follow the present cohort for three years.

## Figures and Tables

**Table 1 ijerph-16-00521-t001:** Demographic characteristics among shift and day workers (N = 94) participating in the study.

Variables	Shift Workers Plant A (N = 42)			Shift Workers Plant B (N = 23)			Day Workers Plant B (N = 29)		
	Number	Mean	SD	Number	Mean	SD	Number	Mean	SD
Age (years)		40.5	11.0		40.0	12.5		49.7	8.5
Women	5			1			7		
BMI (kg/m^2^)		27.2	5.2		26.4	4.4		28.2	4.0
Pack-years		9.2	14.4		5.0	8.0		8.4	10.2
Daily smokers	11			3			2		
College/University	2			1			9		
No. of years as shift worker		14.5	10.1		15.0	9.9		8.3	7.3 *
PA, high intensity (min)		90.1	145		67.0	69.0		121.3	136 *

Adjusted for age; significantly different from all shift workers *, *p* < 0.05. PA: Physical activity.

**Table 2 ijerph-16-00521-t002:** The current 5-week shift plan for Plant A and B. E: Evening shift lasting 8 h, N: Night shift lasting 8 h (N+ = 12 h), D: Dayshift lasting 8 h (D+ =12 h); The working hours for D are 7 a.m. to 3 p.m., D+ are 7 a.m.to 7 p.m., A are 3 p.m. to 11 p.m., N are 11 p.m. to 7 a.m. and N+ are 7 p.m. is 7 a.m.

	Week 1							Week 2							Week 3							Week 4							Week 5						
Weekday	1	2	3	4	5	6	7	1	2	3	4	5	6	7	1	2	3	4	5	6	7	1	2	3	4	5	6	7	1	2	3	4	5	6	7
Plant A shift (N = 42)			E	E	E			N	N				D+	D+	E	E	N	N				D	D	D	D	D							N	N+	N+
Plant A shift (N = 23)	D	D	D	D					N	N	N	N					E	E		N+	N+	N+				D+	D+	D+	E	E					
Plant B day (N = 29)	D	D	D	D	D			D	D	D	D	D			D	D	D	D	D			D	D	D	D	D			D	D	D	D	D		

**Table 3 ijerph-16-00521-t003:** Cardiovascular outcomes among shift and day workers (N = 94).

Variables	Shift Workers Plant A (N = 42)			Shift Workers Plant B (N = 23)			Day Workers Plant B (N = 29)		
	Number	Mean	SD	Number	Mean	SD	Number	Mean	SD
Systolic BP *(mmHg)		121.1	13.9		128.8	25.7		133.7	19.0
Diastolic BP (mmHg)		80.2	7.9		81.8	12.0		84.8	8.3
Supine RHR (bpm)		60.8	10.3		63.3	9.4		60.4	8.6
Supine systolic BP (mmHg)		123.3	11.5		129.2	18.7		134.8	18.4
Supine diastolic BP (mmHg)		72.4	8.2		76.2	14.9		79.2	10.5
Systolic aorta pressure (mmHg)		111.3	10.7		115.9	18.4		123.3	16.5
Diastolic aorta pressure (mmHg)		72.5	10.3		77.3	14.8		80.0	10.6
Plaques ^a^	7			7			8		
Max IMT (mm) ^a^		1.30	0.6		1.32	0.7		1.31	0.4
Augmentation pressure (mmHg)		9.8	3.9		8.9	6.0		14.1	6.5 *
Pulse pressure (mmHg)		37.9	6.0		38.7	6.0		43.3	9.9
Pulse wave velocity (m/s) ^b^		7.6	1.3		7.6	1.3		8.9	1.7 *
CRP (mg/L) ^c^		1.9	2.5		1.2	1.5		1.7	1.5
Cholesterol (mmol/L) ^c^		4.9	0.9		4.5	0.6		4.9	1.0
HDL (mmol/L) ^c^		1.2	0.3		1.1	0.2		1.3	0.3
LDL (mmol/L) ^c^		3.1	0.8		2.8	0.6		3.0	0.9
HbA1c (mmol/mol) ^d^		34.2	3.7		34.6	7.4		37.5	8.2

Adjusted for age, gender and pack-years; significantly different from all shift workers *, *p* < 0.05. ^a^ N = 92. ^b^ N = 88. ^c^ N = 91. ^d^ N = 90. * Blood pressure; RHR: resting heart rate; IMT: intima media thickness; CRP: C-reactive protein; HDL: high density lipoprotein; LDL: low density lipoprotein; HbA1c: glycosylated hemoglobin.

**Table 4 ijerph-16-00521-t004:** The effect of number of years as shift worker on cardiovascular disease (CVD) risk factors.

CVD Outcome Measure	B	*p*
Plaques	0.04	0.20
Max IMT (mm) ^a^	0.015	0.009
Supine RHR (bpm)	0.09	0.45
Supine sBP (mmHg)	−0.13	0.48
Supine dBP(mmHg)	−0.03	0.80
Augmentation pressure (mmHg)	−0.10	0.10
Pulse pressure (mmHg)	−0.11	0.22
A1x (%)	−0.16	0.17
A1 × 75 (%)	−0.15	0.26
PWV * (m/s) ^b^	−0.02	0.33
CRP (mg/L) ^c^	0.06	0.03
HDL (mmol/L) ^c^	0.0007	0.85
LDL (mmol/L) ^c^	0.004	0.66
HbA1c (mmol/mol) ^d^	0.11	0.12

Adjusted for age, sex and pack-years. ^a^ N = 93; ^b^ N = 88; ^c^ N = 91; ^d^ N = 90. * pulse wave velocity; PWV: pulse wave velocity; CRP: C-reactive protein; HDL: high density lipoprotein; LDL: low density lipoprotein; HbA1c: glycosylated hemoglobin; IMT: intima media thickness; RHR: resting heart rate.

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
