# Peer review of "Shift Work Including Night Work and Long Working Hours in Industrial Plants Increases the Risk of Atherosclerosis"

_ijerph, 2019, doi:10.3390/ijerph16030521_

Round 1
Reviewer 1 Report
Overall this is a well-designed study and a nicely written manuscript. I have several concerns that need to be addressed, and suggestions for improving the manuscript below.
Please explain in greater detail the “5-shift plan” schedule. I suggest a table or diagram might help clarify. At the moment, it is unclear whether shift workers include day, evening, and night shift, whether all shift workers were on rotation, how long rotations were etc.
When referring to the day shift in the Study Design and Population section – does this refer to the “day workers”? Or to day shifts within shift work rotations? Need to be careful to keep language consistent. If this does refer to day workers, then surely, they are also shift workers, and you are comparing day shift work to rotating shift work? Please clarify the exact nature of the shifts, as this is critically important for study outcomes.
Some information on the nature of the industrial work would be useful. Is there any exposure to toxins? Exposure to anything that would increase the risk of CVD? If so, did this differ between the plants and between day workers vs shift workers?
Were behaviors related to CVD (alcohol intake, smoking, unhealthy eating habits, sleep etc.) not measured? If not, why not given their established relationship? It seems like these would be important covariates.
Was any attrition observed or did all participants complete all aspects of the study?
Some information on study procedures would be useful. For example, when did participants get tested (during the shift, immediately after a shift, during time off)? Where did participants get tested (at the work site or some other site)? How long did testing take?
The finding that day workers had higher augmented pressure and pulse wave velocity needs describing – it seems this is not a consequence of age, so it is counter to what would be expected? Please explain.
Author Response
Overall this is a well-designed study and a nicely written manuscript. I have several concerns that need to be addressed, and suggestions for improving the manuscript below.
Please explain in greater detail the “5-shift plan” schedule. I suggest a table or diagram might help clarify. At the moment, it is unclear whether shift workers include day, evening, and night shift, whether all shift workers were on rotation, how long rotations were etc.
We have now included a figure (Figure 1) in the paper which better describes the 5-shift plan including rotation and duration of the shift work.
When referring to the day shift in the Study Design and Population section – does this refer to the “day workers”? Or to day shifts within shift work rotations? Need to be careful to keep language consistent. If this does refer to day workers, then surely, they are also shift workers, and you are comparing day shift work to rotating shift work? Please clarify the exact nature of the shifts, as this is critically important for study outcomes.
“Day workers” are now used for all those who work only during day time. They are compared to those who work rotating shifts- the “shift workers”.
Some information on the nature of the industrial work would be useful. Is there any exposure to toxins? Exposure to anything that would increase the risk of CVD? If so, did this differ between the plants and between day workers vs shift workers?
Since the production is mostly automatized, the work is mostly sedentary both for those who work rotating shifts or those working only during daytime in these plants producing insulation materials. There could be exposures of dust – even silicate- but this is seldom. Most potential dust exposure is of big particles that would deposit in the upper airways. These potential exposures are similar in the two plants. There could be exposures to aerosols such as ammonia but this gas dissolves mostly in the upper airways. The potential exposures are also present for those who only work during daytime. According to the occupational health services, these exposures are of low levels and not considered giving increased risk for CVD. This information is now included in the manuscript in “methods”
Were behaviors related to CVD (alcohol intake, smoking, unhealthy eating habits, sleep etc.) not measured? If not, why not given their established relationship? It seems like these would be important covariates.
We recorded smoking habits and history but not eating habits nor alcohol intake. As for alcohol habits, mostly the literature states that rotating shift workers are consuming less alcohol than day workers and this was also the impression after talking to the workers. However, these habits were not systematically registered since we considered it sensitive information and we wanted to establish a close relationship with the cohort since they will be subjected to a long follow-up. This is now included in the limitations in the discussion part of the paper. The weight and height were measured and the shift workers were generally not obese- see table 1. As for sleep, we are planning a follow-up of this with daily registration for a period of some weeks also including registration by accelerometer.
Was any attrition observed or did all participants complete all aspects of the study?
All participants completed all aspects of the study.
Some information on study procedures would be useful. For example, when did participants get tested (during the shift, immediately after a shift, during time off)? Where did participants get tested (at the work site or some other site)? How long did testing take?
The participants were tested during daytime; mostly during dayshift or when they were off duty. All tests, except for the examination at the cardiologist which was in the same building, were performed at the occupational health services which was a 45 min drive form plant A and a 5 minute drive for those coming from plat B. The tests took all together between 45 and 60 minutes per person. This information is now provided in “Methods”
The finding that day workers had higher augmented pressure and pulse wave velocity needs describing – it seems this is not a consequence of age, so it is counter to what would be expected? Please explain.
We controlled for age but still augmentation pressure and PWV were increased among day workers. We have no good explanation for this but the phenomenon could be due to selection mechanisms; the “healthy worker effect”. One could be tempted to consider the shift worker more healthy than dayworkers. Since they choose to work nightshifts their health is probably initially good. The healthy worker effect is discussed under limitations in the discussion part of the paper.

Reviewer 2 Report
Thank you for this important work examining the association between shift work and cardiovascular disease. While the period of follow up for 3 years is preliminary, your findings suggest there is a link between shift work and the development of disease. You have used a broad range of measurements to characterise the progression of potential cardiovascular disease, a battery that is novel in this group.
My main suggestions would be to revise the abstract. There are some aspects that seem unfinished, especially in the conclusions.
I would also suggest the addition of a limitations section in your discussion. The short follow up and large potential for confounding needs to be addressed given the broad range of influencing factors that could lead to cardiovacscular disease. It would be helpful to address these or acknowledge those that may have influenced the results.
Thank you for this important piece of work that will help to better characterise the onset of cardiovascular disease in this vulnerable group of workers.
Author Response
Thank you for this important work examining the association between shift work and cardiovascular disease. While the period of follow up for 3 years is preliminary, your findings suggest there is a link between shift work and the development of disease. You have used a broad range of measurements to characterise the progression of potential cardiovascular disease, a battery that is novel in this group.
My main suggestions would be to revise the abstract. There are some aspects that seem unfinished, especially in the conclusions.
We have revised the abstract- and we hope it has improved.
I would also suggest the addition of a limitations section in your discussion. The short follow up and large potential for confounding needs to be addressed given the broad range of influencing factors that could lead to cardiovacscular disease. It would be helpful to address these or acknowledge those that may have influenced the results.
Strengths and limitations of the study is included in the discussion part.
Thank you for this important piece of work that will help to better characterise the onset of cardiovascular disease in this vulnerable group of workers.